# Deep inspiration breath hold versus free breathing in postoperative radiotherapy strategy for patients with left-sided breast cancer treated with volumetric modulated arc therapy: A meta-analysis and systematic review

**Pin-Yi Chiang[1,2◉], Pin-Jui Huang[2,3◉], Chao-Hsiung Hung[4], Ching-Po Lin[2,5,6◉]\*, Chih-Chia Chang [1,7◉]\***

1 Department of Radiation Oncology, Ditmanson Medical Foundation Chia-Yi Christian Hospital, Chiayi City, Taiwan, 2 Ph.D. Program of Interdisciplinary Medicine, National Yang Ming Chiao Tung University, Taipei City, Taiwan, 3 Division of Urology, Department of Surgery, Dajia Lee Hospital, Taichung City, Taiwan, 4 Department of Radiation Oncology, Chiayi Chang Gung Memorial Hospital, Chiayi County, Taiwan, 5 Institute of Neuroscience, National Yang Ming Chiao Tung University, Taipei City, Taiwan, 6 Department of Education and Research, Taipei City Hospital, Taipei City, Taiwan, 7 Department of Nursing, Chung-Jen Junior College of Nursing, Health Sciences and Management, Chiayi County, Taiwan

◉ These authors contributed equally to this work.
\* 07229@cych.org.tw (CCC); chingpolin@gmail.com (CPL)

## Abstract

This meta-analysis aimed to determine the effect of deep inspiration breath hold (DIBH) compared with free breathing (FB) on dose to the organs at risk (OARs), such as the heart, left anterior descending (LAD) coronary artery, lungs, and contralateral breast, in patients with left-sided breast cancer treated with volumetric modulated arc therapy (VMAT). PubMed, EMBASE, and Cochrane Library electronic databases were searched for studies until March 21, 2024. Cochrane RevMan version 5.4 statistical software was used to analyze 11 eligible studies. Standard mean difference (SMD), with 95% confidence interval for OAR dose reductions, was calculated. DIBH considerably resulted in lower mean doses (Dmean) to the heart (SMD = −1.40 Gy), LAD (SMD = −1.65 Gy), ipsilateral lung (SMD = −0.57 Gy), contralateral lung (SMD = −0.46 Gy), and contralateral breast (SMD = −0.20 Gy). If VMAT was delivered with an arc of >180ᴼ, the heart Dmean reduction was even more pronounced. Subgroup analysis revealed that DIBH efficiently reduced heart Dmean, especially in patients with tumor bed boost without nodal irradiation. DIBH was effective in reducing dose to OARs in patients treated with VMAT in all subgroups, i.e., breast only, with/without tumor bed boost, and with/without nodal irradiation. Furthermore, the use of DIBH is strongly recommended for patients undergoing VMAT with a tumor bed boost or without nodal irradiation, as it is more effective in reducing heart Dmean than FB.

**Data availability statement:** All relevant data are within the paper.

**Funding:** This study was supported by Ditmanson Medical Foundation Chia-Yi Christian Hospital Research Program (R114-060) awarded to P.C.

**Competing interests:** The authors declare that they have no known competing financial interests or personal relationships that could have appeared to influence the work reported in this paper.

## Introduction

Breast cancer is a prevalent malignancy in women, and postoperative radiotherapy (RT) is considered the standard treatment for early-stage cases as it significantly reduces local recurrence and improves long-term survival rates [1–4]. However, this treatment modality poses risks, as breast irradiation causes substantial radiation exposure to the heart and ipsilateral lung. Notably, left-sided breast cancer (LSBC) irradiation presents an increased risk of cardiac morbidity and mortality due to the proximity of critical anterior cardiac structures, such as the left anterior descending coronary artery (LAD), potentially causing abnormal cardiac perfusion post-irradiation [5–8]. Darby et al. revealed that a 1 Gy increase in heart mean dose (Dmean) is correlated with a 7.4% increase in the relative risk for major coronary events, highlighting the importance of reducing radiation exposure to mitigate the risk of ischemic heart disease in patients with breast cancer [6].

Efforts to develop techniques that minimize radiation doses to organs at risk (OARs), particularly the heart and lungs, are ongoing. However, currently available RT methods cause incidental cardiac irradiation. Therefore, optimal techniques are required for reducing heart and ipsilateral lung irradiation while maintaining target coverage.

Deep inspiration breath hold (DIBH) is one of the treatment strategies for LSBC. In this technique, patients are instructed to take a deep breath and hold it during radiation delivery, causing lung expansion and heart displacement from the chest wall, thereby increasing the distance between the heart and the treatment area. Specialized systems are used to monitor and maintain the breath hold position throughout treatment, ensuring accuracy. Various devices, such as the real-time position management system and active breathing control, facilitate DIBH, thereby demonstrating favorable feasibility and reproducibility in reducing irradiated heart volume and heart Dmean [9–12]. Even without any device, voluntary deep inspiration breath hold has proven to be feasible and accurate, similar to free breathing (FB) [13].

Although DIBH improves dosimetric outcomes, it requires patient cooperation and may prolong treatment due to additional setup and monitoring. In addition to DIBH, advanced techniques such as intensity-modulated RT (IMRT) and volumetric modulated arc therapy (VMAT) provide precise radiation delivery while minimizing OAR exposure [14–17]. VMAT, an advanced form of IMRT, delivers radiation through continuous gantry rotation while simultaneously adjusting dose rate, gantry speed, and multileaf collimator (MLC) positions. This enables highly conformal dose distribution, shorter delivery times, and fewer monitor units [18]. In some situations involving complex anatomy—such as comprehensive nodal irradiation including the internal mammary chain—VMAT may offer more favorable sparing of cardiac substructures and lungs from high-dose radiation exposure. However, it is also recognized that VMAT and IMRT generally deliver greater low-dose exposure to surrounding normal tissues, including the heart, lungs, and contralateral breast, compared to three-dimensional conformal RT (3D-CRT). 3D-CRT has long been the conventional standard in breast radiotherapy, typically using fixed tangential fields to irradiate the tumor bed or chest wall while minimizing exposure to adjacent healthy tissues. The rotational nature of VMAT, while improving conformity, may also result in multiple beam paths through non-target tissues. This may potentially offset some of the cardiac-sparing benefits

achieved by DIBH and may lead to increased radiation exposure to other OARs—such as the lungs and contralateral breast—due to the inherently increased low-dose exposure associated with VMAT.

Therefore, this meta-analysis aims to evaluate whether the use of DIBH, compared with FB, in VMAT-based postoperative RT for LSBC effectively reduces OAR doses, and to assess the trade-off between high-dose sparing and low-dose exposure associated with VMAT. Previous meta-analyses have examined the dosimetric benefits of DIBH versus FB for OARs in breast cancer; however, they pooled data from studies using various radiotherapy techniques including 3D-CRT, IMRT, and VMAT [19,20], which may confound the interpretation of technique-specific effects. To the best of our knowledge, the present study is the first meta-analysis specifically focused on VMAT combined with DIBH versus FB. We analyzed the impact of this combination on OARs, including the heart, LAD, both lungs, and contralateral breast. Inspiration for this study was drawn from a retrospective single-institution study by Razvi et al. [21], which showed that wide tangents (including internal mammary chain), boost treatments, smaller heart volume, and chest wall irradiation were associated with higher heart Dmean in patients treated with tangential fields. Building on these insights, we conducted subgroup analyses to evaluate whether DIBH continues to reduce heart Dmean in VMAT plans across various clinical settings, including different target volumes and the presence or absence of a tumor bed boost. This was done to assess whether the benefit of DIBH is consistent and potentially applicable across different patient conditions. Furthermore, we investigated whether restricted or extensive partial arc angles more effectively reduce the heart Dmean. The results of this study are expected to provide an evidence-based framework for optimizing treatment outcomes and minimizing adverse effects in patients with breast cancer undergoing VMAT-based RT.

## Materials and methods

### Search strategy

A thorough literature review was conducted to investigate the clinical dose and efficacy of postoperative radiotherapy (RT) using deep inspiration breath hold (DIBH) versus free breathing (FB). Relevant studies were identified through a systematic search of PubMed, EMBASE, and the Cochrane Library on March 21, 2024. The literature selection process adhered to the Preferred Reporting Items for Systematic Reviews and Meta-Analyses (PRISMA) guidelines and the Assessing the Methodological Quality of Systematic Reviews (AMSTAR) criteria. The primary endpoint was the reduction in mean heart dose (Dmean), while secondary endpoints included reductions in the ipsilateral lung Dmean and left anterior descending coronary artery (LAD) Dmean.

To ensure transparency and reproducibility, the detailed search strategies employed for PubMed, the Cochrane Library, and Embase have been provided as a supplementary file. The search terms combined controlled vocabulary (MeSH/Emtree terms) and relevant keywords to enhance both the precision and sensitivity of the search. A summary of the search logic is as follows:

- **PubMed**: ("Breath Holding"[MeSH Terms] OR "deep inspiration breath hold" OR DIBH) AND ("Respiration"[MeSH Terms] OR "free breathing") AND ("Breast Neoplasms"[MeSH Terms] OR "left-sided breast cancer")

- **Cochrane Library**: ("deep inspiration breath hold" OR DIBH) AND ("free breathing" OR "normal breathing") AND ("left-sided breast" OR "breast cancer")

- **EMBASE**: ('breath holding'/exp OR 'deep inspiration breath hold' OR DIBH) AND ('normal breathing'/exp OR 'free breathing') AND ('breast cancer'/exp OR 'left-sided breast')

### Inclusion and exclusion criteria

Studies that met the following criteria were included:

(1) studies that compared the heart Dmean of DIBH with that of FB;

(2) studies with adult participants diagnosed with LSBC;

(3) studies in which patients were treated with VMAT;

(4) studies with no statistically significant differences in the basic characteristics of the participants;

(5) studies that included at least one of the following outcomes: heart V5/V25, LAD Dmean/Dmax, ipsilateral lung Dmean/V20, contralateral lung Dmean, and contralateral breast Dmean.

The exclusion criteria were as follows:

(1) performed RT for bilateral breast or only right-sided breast;

(2) non-English literature;

(3) publication types such as conference articles, letters, comments, and reviews.

## Data extraction and quality assessment

Two reviewers (PYC and CHH) independently searched and extracted data from the literature following the inclusion criteria. The extracted information included author, publication year, country, study design, intervention, sample size, follow-up duration, and outcomes. Discrepancies were resolved through discussions among all authors. We used parameters such as the mean dose (Dmean); maximum dose (Dmax); and percentage of the organ volume receiving at least 5 Gy (V5), 20 Gy (V20), and 30 Gy (V30) to analyze dose distributions for the heart, LAD, and left lung. Two reviewers (PYC and PJH) independently conducted the quality assessment and data extraction using a standardized form. Each reviewer worked separately to minimize bias and enhance objectivity. In the event of discrepancies between the two reviewers, disagreements were first discussed to reach a consensus. If consensus could not be achieved, a third reviewer (CCC) was consulted to resolve the conflict and make the final decision. This approach ensured the reliability and consistency of the quality assessment process.

The Newcastle–Ottawa Scale (NOS) was used to assess the quality of nonrandomized studies based on three broad perspectives: study group selection, group comparability, and exposure or outcome of interest ascertainment for case-control or cohort studies, respectively.

## Statistical analysis

Cochrane Q and $I^2$ statistics were used to assess the heterogeneity across studies. $P$-values of >0.10 and $I^2$ values of <50% indicated no heterogeneity among the included studies. A random-effects model (REM) was applied throughout all parameters. Standard mean difference (SMD) and 95% confidence interval (CI) were used to analyze the effects of measurement data. $P$-values of <0.05 were considered statistically significant. In addition, we conducted sensitivity analyses to explore the sources of substantial heterogeneity among the included studies. Five distinct approaches were employed to identify potential contributors to this heterogeneity:

1. **Exclusion of high-risk studies**: Studies assessed as having a high risk of bias were excluded to evaluate their impact on the overall pooled estimates.

2. **Alternative statistical models**: Both fixed-effect and random-effects models were applied to assess whether the choice of model influenced the meta-analytic results.

3. **Leave-one-out analysis**: Each study was sequentially excluded to determine whether any single study had a disproportionate influence on the overall findings.

4. **Stricter inclusion criteria**: Sensitivity analyses were also performed by restricting the dataset to randomized controlled trials (RCTs) only, to assess whether study design affected the results.

5. **Subgroup-specific exclusions**: Studies from specific populations (e.g., particular geographic regions) were excluded to investigate whether these subgroups contributed to the observed heterogeneity.

Moreover, funnel plot was used to understand the bias of literature publication. The possibility of publication bias was low if the points in the funnel plot were symmetrically distributed on both sides around the middle dashed line and concentrated in the middle. Otherwise, the possibility of publication bias was considered high. The Cochrane RevMan version 5.4 software was used for all statistical analyses.

## Results

### Study selection and characteristics of the included studies

After screening for relevant studies, 11 studies with 217 patients were included in this analysis (Table 1) [22–32]. A flowchart detailing the study selection process is presented in the form of a 2020 PRISMA flowchart (Fig 1). Table 2 outlines the baseline characteristics of the included studies, including breast cancer pathology, mean age, treatment type, prescribed dose, tumor bed boost status, arc angle, and lymph node involvement. Table 3 summarizes the dose–volume metrics to OARs per study, and Table 4 presents the standard mean difference (SMD) between DIBH and FB over all studies.

### Risk of bias in the included studies

Overall, all studies were dosimetric comparisons and mentioned no follow-up plan due to their design (Table 5). The adequacy of the follow-up of cohorts could not be estimated. All studies present good quality in terms of exposed cohort, selection of nonexposed cohort, ascertainment of exposure, comparability of cohorts based on the design or analysis, and assessment of outcome. The total score of each study represents adequate power for the test of risk of bias.

### Effects of DIBH and FB on heart dose

The DIBH group showed significant differences and lower radiation exposure in heart Dmean (SMD = −1.40 Gy; 95% CI = −1.69 to −1.10 Gy; $I^2$ = 35%; $p < 0.01$; Fig 2a), heart V5 (SMD = −1.12 Gy; 95% CI = −1.40 to −0.84 Gy; $I^2$ = 6%; $p < 0.01$; Fig 2b), heart V10 (SMD = −1.13 Gy; 95% CI = −1.43 to −0.82 Gy; $I^2$ = 0%; $p < 0.01$; Fig 2c), heart V20 (SMD = −1.09 Gy; 95% CI = −1.44 to −0.74 Gy; $I^2$ = 0%; $p < 0.01$; Fig 2d), heart V25 (SMD = −1.11 Gy; 95% CI = −1.62

**Table 1. Summary of comparative studies included in meta-analysis.**

| Study | Country | Study Period | Study design | LE | Intervention | | Sample size | |
|---|---|---|---|---|---|---|---|---|
| | | | | | Trial | Control | Trial | Control |
| **Osman 2014 [22]** | Netherlands | 2012.07-2013.11 | case series | 5★ | DIBH | FB | 13 | 13 |
| **Swamy 2014 [23]** | India | NR | case series | 5★ | DIBH | FB | 10 | 10 |
| **Pham 2016 [24]** | Australia | 2011.01-2013.07 | case series | 5★ | DIBH | FB | 15 | 15 |
| **Sakka 2017 [25]** | Germany | 2015.01-2015.05 | case series | 5★ | DIBH | FB | 20 | 20 |
| **Dumane 2018 [26]** | USA | NR | case series | 5★ | DIBH | FB | 10 | 10 |
| **Kuo 2019 [27]** | USA | NR | case control | 5★ | DIBH | FB | 10 | 10 |
| **Yeh 2019 [28]** | Taiwan | 2015.06-2016.09 | case series | 5★ | DIBH | FB | 12 | 12 |
| **Zhang 2020 [29]** | China | 2018.01-2019.06 | case series | 5★ | DIBH | FB | 48 | 48 |
| **Chen 2020 [30]** | China | 2018.03-2019.06 | case series | 5★ | DIBH | FB | 19 | 19 |
| **Cheung 2022 [31]** | Hong Kong | 2017.01-2020.10 | case series | 5★ | DIBH | FB | 15 | 15 |
| **Xu 2023 [32]** | China | 2020.01-2021.09 | case control | 5★ | DIBH | FB | 18 | 27 |

LE = Level of evidence of New-castle Ottawa Scale, NR = Not Reported.

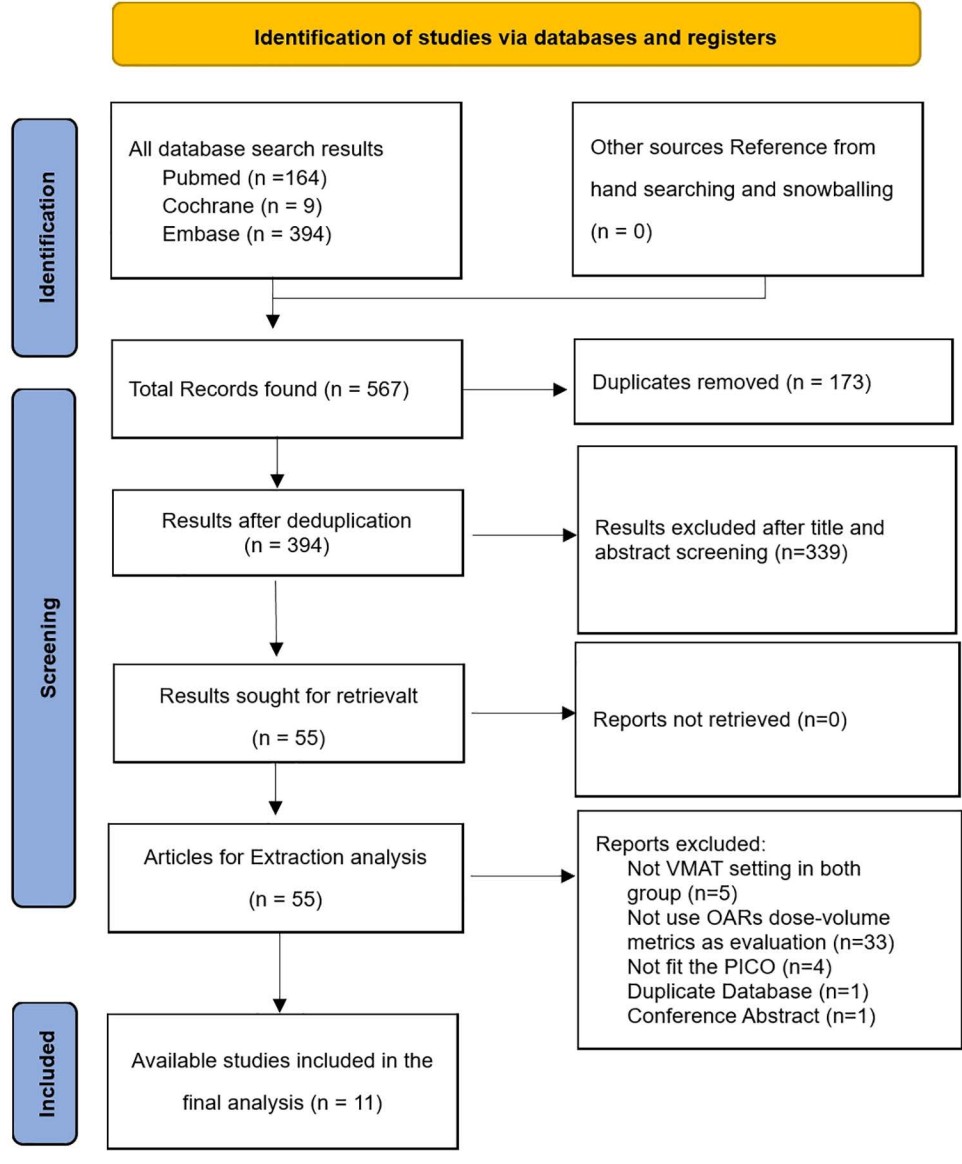

**Fig 1. 2020 PRISMA flowchart.**

to −0.60 Gy; I²=0%; $p < 0.01$; Fig 2e), and heart V30 (SMD = −0.93 Gy; 95% CI = −1.31 to −0.55 Gy; I²=0%; $p < 0.01$; Fig 2f). Moreover, the DIBH group showed smaller heart volume (SMD = −0.58 Gy; 95% CI = −0.96 to −0.21 Gy; I²=0%; $p < 0.01$; Fig 2g).

## Effects of DIBH and FB on LAD dose

The DIBH group demonstrated significant differences and lower radiation exposure in LAD Dmax (SMD = −0.87 Gy; 95% CI = −1.21 to −0.54 Gy; I²=15%; $p < 0.01$; Fig 3a). Similar results were observed for LAD Dmean, albeit with higher heterogeneity (SMD = −1.65 Gy; 95% CI = −2.17 to −1.13 Gy; I²=69%; $p < 0.01$; Fig 3b), as confirmed through sensitivity analysis.

**Table 2. Baseline characteristics of included studies.**

| Study | Sample size | Age (years) | Stage (pts) | DIBH device | RTP | Prescription dose | Dose of tumor bed boost | Arc angle | Treated area |
|---|---|---|---|---|---|---|---|---|---|
| Osman 2014 | 13 | mean 57 | NR | RPM | Eclipse | 42.56 Gy/16 | N/A | two partial arcs(210°-250° per arc) | breast/CW + IMN + SCF |
| Swamy 2014 | 10 | NR | III | RPM | Eclipse | 50Gy/25 | N/A | two partial arcs(starting gantry angle from 179° to stop angle 300°) | CW + SCF |
| Pham 2016 | 15 | NR | NR | RPM | Eclipse | 50Gy/25 | N/A | two partial arcs (starting gantry angle: between 295°-315°; stop angle between 125°-145°) | breast + IMN + SCF + axilla |
| Sakka 2017 | 20 | <70 | NR | RPM | Eclipse | 50.4Gy/28 | N/A | four partial arcs (~240° per arc) | breast |
| Dumane 2018 | 10 | NR | II–III | RPM | Eclipse | 50Gy/25 | N/A | two partial arcs(190°-220° per arc) | CW + implant + IMN + SCF + axilla |
| Kuo 2019 | 20 | NR | NR | RPM | Eclipse | 50Gy/25 | N/A | four or five partial arcs(100°-120° per arc) | CW + implant + IMN + SCF + axilla |
| Yeh 2019 | 12 | mean 50.2 (38-69) | Tis(4), I(4), II(1), III(3) | Abches | Pinnacle | 50Gy/25 | 10Gy/5 | two to four partial arcs(~45° per arc) | breast/CW: 9 pts breast + IMN + SCF + axilla: 3 pts |
| Zhang 2020 | 48 | <70 | NR | AlignRT | Pinnacle | 50Gy/25 | 60Gy/25 (SIB) | three or four partial arcs (starting gantry angle: between 310° and 320°; stop angle between 135° −150°) | breast |
| Chen 2020 | 19 | median 52 (31-69) | III (≥N2) | RPM | Eclipse | 50Gy/25 | N/A | NR | CW + IMN + SCF |
| Cheung 2022 | 15 | NR | NR | RPM | Eclipse | 50.4Gy/28 | 58.8Gy/28 (SIB) | two partial arcs(230°-240° per arc) | breast: 11 pts breast + SCF: 4 pts |
| Xu 2023 | 45 | 48.89 ± 8.0 (mean ± std) | Tis(7), I(31), II(7) | RPM | Eclipse | 42.56Gy/16 | 10Gy/5 | two partial arcs (240° per arc) | breast |

NR = Not Reported, std = standard deviation.

## Effects of DIBH and FB on ipsilateral and contralateral lung dose

The DIBH group demonstrated significant differences and lower radiation exposure with high heterogeneity in ipsilateral lung Dmean (SMD = −0.57 Gy; 95% CI = −0.93 to −0.20 Gy; I²=61%; $p<0.01$; Fig 4a) and ipsilateral lung V10 (SMD = −0.62 Gy; 95% CI = −1.14 to −0.10 Gy; I²=68%; $p=0.02$; Fig 4b). Sensitivity analysis confirmed these results. Ipsilateral lung V20 data indicated significant differences and lower radiation exposure with low heterogeneity (SMD = −0.57 Gy; 95% CI = −0.84 to −0.30 Gy; I²=21%; $p<0.01$; Fig 4c).

However, ipsilateral lung V5 did not yield statistically significant results (SMD = −0.58 Gy; 95% CI = −1.16 to −0.01 Gy; I²=78%; $p=0.05$; Fig 4d). A smaller statistically significant ipsilateral lung volume was observed (SMD = 1.99 Gy; 95% CI = 1.53 to 2.46 Gy; I²=29%; $p<0.01$; Fig 4e).

Furthermore, the DIBH group demonstrated significant differences and lower radiation exposure in contralateral lung Dmean (SMD = −0.46 Gy; 95% CI = −0.70 to −0.22 Gy; I²=0%; $p<0.01$; Fig 4f) and contralateral lung V5 (SMD = −0.85 Gy; 95% CI = −1.24 to −0.46 Gy; I²=50%; $p<0.01$; Fig 4g). However, contralateral lung V20 did not yield statistically significant results (SMD = −0.42 Gy; 95% CI = −0.96 to 0.12 Gy; I²=0%; $p=0.13$; Fig 4h).

**Table 3. Dose–volume metrics to OARs per study.**

| Study | Breath | Case number | Heart Dmean (Gy) | Heart V5 (%) | Heart V25 (%) | LAD Dmean (Gy) | LAD Dmax (Gy) | Ips. Lung Dmean (Gy) | Ips. Lung V20 (%) | Cont. Lung Dmean (Gy) | Cont. Breast Dmean (Gy) |
|---|---|---|---|---|---|---|---|---|---|---|---|
| Osman 2014 | DIBH/ FB | 13 | D: 4.1±1.4 F: 5.8±1.6 | D: 18.7±11.1 F: 33.6±16.3 | N/A | N/A | N/A | D:13.3±3.1 F: 14.0±3.4 | D:26.5±8.9 F:27.9±11.5 | D:2.6±0.6 F:3.4±1.2 | D:2.5±1.0 F:2.8±0.9 |
| Swamy 2014 | DIBH/ FB | 10 | D: 9.9±2.1 F: 12.0±2.9 | N/A | N/A | N/A | N/A | D:11.7±1.6 F:13.0±1.6 | D:18.6±2.9 F:21.7±3.9 | D:4.4±0.7 F:5.4±0.8 | D:5.6±1.2 F:5.7±1.1 |
| Pham 2016 | DIBH/ FB | 15 | D: 5.7±1.4 F: 8.1±2.0 | D: 35.8±14.6 F: 50.4±13.5 | D: 1.9±1.8 F:5.9±4.1 | D:17.4±5.8 F:24.7±6.5 | D:33.3±8.9 F:40.9±6.0 | D:18.2±0.9 F:18.2±1.1 | D:34.3±3.3 F:35.3±3.4 | D:4.7±0.7 F:6.0±1.1 | D:5.0±1.0 F:5.1±1.0 |
| Sakka 2017 | DIBH/ FB | 20 | D: 4.0±0.7 F: 5.3±1.1 | N/A | N/A | D:7.3±1.0 F:8.7±1.8 | D:15.5±5.6 F:21.2±6.3 | D:9.9±1.0 F:10.3±1.1 | N/A | D:3.5±0.6 F:4.3±0.8 | D:3.3±0.6 F:3.6±0.9 |
| Dumane 2018 | DIBH/ FB | 10 | D: 5.3±1.0 F: 8.2±1.4 | D: 35.3±9.9 F: 64.9±14.2 | D: 1.0±1.5 F: 3.1±2.1 | N/A | D:30.8±12.6 F:40.7 12.1 | D:14.9±1.6 F:15.7±1.8 | D:26.4±3.9 F:28.8±2.5 | N/A | D:5.1±1.4 F:5.7±1.4 |
| Kuo 2019 | DIBH/ FB | DIBH: 10 FB:10 | D: 6.6±0.8 F: 7.5±1.1 | N/A | D: 1.8±1.0 F: 3.5±2.2 | N/A | D:31.4±7.3 F:34.0±11.5 | D:15.9±1.1 F:16.1±1.2 | D:27.5±3.4 F:28.8±2.5 | N/A | N/A |
| Yeh 2019 | DIBH/ FB | 12 | D: 4.2±2.1 F: 6.7±2.6 | D: 14.4±8.3 F:22.7±10.8 | N/A | N/A | N/A | D:10.1±3.4 F:11.5±4.0 | D:19.5±7.8 F:22.9±6.7 | D:0.7±0.5 F:0.9±0.6 | N/A |
| Zhang 2020 | DIBH/ FB | 48 | D: 3.6±0.9 F: 5.4±1.6 | D:15.9±9.0 F:24.1±8.6 | N/A | D:3.9±1.1 F:6.9±1.8 | N/A | D:9.5±1.3 F:11.3±1.3 | D:16.5±2.6 F:19.5±3.0 | D:1.6±0.6 F:2.1±1.1 | D:2.1±0.7 F:2.6±1.0 |
| Chen 2020 | DIBH/ FB | 19 | D: 4.8±0.6 F: 6.3±1.2 | D:24.5±4.9 F:37.0±10.4 | N/A | D:14.2±4.0 F:18.7±3.7 | N/A | D:14.8±0.6 F:14.9±0.7 | D:28.1±1.7 F:28.4±2.2 | D:4.9±1.6 F:5.2±1.5 | D:4.6±2.0 F:5.8±2.3 |
| Cheung 2022 | DIBH/ FB | 15 | D: 2.6±0.4 F: 6.3±1.2 | D:8.3±3.2 F:12±3.0 | N/A | D:4.9±0.5 F:6.1±0.6 | D:15.2±1.8 F:18.2±1.7 | D:8.8±0.6 F:9.7±0.6 | D:13.2±1.7 F:14.8±1.7 | D:2.3±0.2 F:2.7±0.1 | D:1.9±0.2 F:2.1±0.2 |
| Xu 2023 | DIBH/ FB | DIBH: 18FB:27 | D: 2.4±0.4 F: 3.9±0.6 | N/A | N/A | D:6.3±1.2 F:9.8±1.4 | D:26.8±6.6 F:30.6±5.1 | N/A | N/A | N/A | N/A |

DIBH = Deep Inspirational Breath Hold, FB = Free Breath, Ips. = Ipsilateral, Cont. = Contralateral.

**Table 4. Statistical results of OAR dose–volume metrics for the DIBH and FB groups.**

| Dose-volume metrics | Study number | Sample size | | Heterogeneity (total) | | | | SMD (95% CI) | P-value(overall) |
|---|---|---|---|---|---|---|---|---|---|
| | | DIBH | FB | Chi² | Df | I²% | P value | | |
| Heart Dmean | 11 | 190 | 199 | 15.45 | 10 | 35% | 0.12 | −1.40 [−1.69, −1.10] | p<0.01 |
| Heart Volume | 3 | 79 | 79 | 2.53 | 2 | 21% | 0.28 | −0.58 [−0.96, −0.21] | p<0.01 |
| Heart V5 | 7 | 132 | 132 | 6.38 | 6 | 6% | 0.38 | −1.12 [−1.40, −0.84] | p<0.01 |
| Heart V10 | 4 | 97 | 97 | 0.87 | 3 | 0% | 0.83 | −1.13 [−1.43, −0.82] | p<0.01 |
| Heart V20 | 5 | 74 | 74 | 1.29 | 4 | 0% | 0.86 | −1.09 [−1.44, −0.74] | p<0.01 |
| Heart V25 | 3 | 35 | 35 | 0.20 | 2 | 0% | 0.91 | −1.11 [−1.62, −0.60] | P<0.01 |
| Heart V30 | 5 | 69 | 59 | 2.51 | 4 | 0% | 0.64 | −0.93 [−1.31, −0.55] | p<0.01 |
| LAD Dmean | 6 | 135 | 144 | 16.10 | 5 | 69% | < 0.01 | −1.65 [−2.17, −1.13] | p<0.01 |
| LAD Dmax | 6 | 88 | 97 | 5.92 | 5 | 15% | 0.31 | −0.87 [−1.21, −0.54] | p<0.01 |
| Ips. Lung Volume | 4 | 92 | 92 | 4.21 | 3 | 29% | 0.24 | 1.99 [1.53, 2.46] | p<0.01 |
| Ips. Lung Dmean | 10 | 172 | 172 | 22.89 | 9 | 61% | <0.01 | −0.57 [−0.93, −0.20] | p<0.01 |
| Ips. Lung V5 | 7 | 132 | 132 | 27.72 | 6 | 78% | <0.01 | −0.58 [−1.16, −0.01] | p=0.05 |
| Ips. Lung V10 | 6 | 112 | 112 | 15.41 | 5 | 68% | <0.01 | −0.62 [−1.14, −0.10] | p=0.02 |
| Ips. Lung V20 | 9 | 152 | 152 | 10.12 | 8 | 21% | 0.26 | −0.57 [−0.84, −0.30] | P<0.01 |
| Cont. Lung Dmean | 8 | 150 | 140 | 2.77 | 7 | 0% | 0.91 | −0.46 [−0.70, −0.22] | p<0.01 |
| Cont. Lung V5 | 7 | 132 | 132 | 12.09 | 6 | 50% | 0.06 | −0.85 [−1.24, −0.46] | p<0.01 |
| Cont. Lung V20 | 4 | 48 | 38 | 1.74 | 2 | 0% | 0.42 | −0.42 [−0.96, 0.12] | p=0.13 |
| Cont. Breast Dmean | 8 | 150 | 140 | 6.44 | 7 | 0% | 0.49 | −0.20 [−0.31, −0.09] | p<0.01 |
| Cont. Breast V5 | 4 | 95 | 95 | 8.90 | 3 | 66% | 0.03 | −0.53 [−1.07, 0.01] | p=0.06 |

CI = confidence interval, SMD = standard mean difference, Ips. = Ipsilateral, Cont. = Contralateral.

### Effects of DIBH and FB on contralateral breast dose

The DIBH group demonstrated significant differences and lower radiation exposure in contralateral breast Dmean (SMD = −0.20 Gy; 95% CI = −0.31 to −0.09 Gy; I²=0%; p<0.01; Fig 5a). Contralateral breast V5 did not yield statistically significant results (SMD = −0.53 Gy; 95% CI = −1.07 to 0.01 Gy; I²=66%; p=0.06; Fig 5b).

### Subgroup analysis of heart Dmean between DIBH and FB

We determined key factors, including tumor bed boost, arc angle, and lymph node irradiation impacting the heart dose reduction in all groups.

The subgroup analysis regarding the use of tumor bed boost indicated statistically significant and low heterogeneity in the group without tumor bed boost (SMD = −1.28 Gy; 95% CI = −1.60 to −0.97 Gy; I²=0%; p<0.01; Fig 6a). In contrast, the group with tumor bed boost demonstrated statistically significant but high heterogeneity (SMD = −1.57 Gy; 95% CI = −2.20 to −0.94 Gy; I²=68%; p<0.01; Fig 6a).

The subgroup analysis based on the degree of arc angle revealed statistically significant and low heterogeneity in the group with an arc angle of <180° (restricted partial arcs) (SMD = −0.96 Gy; 95% CI = −1.59 to −0.33 Gy; I²=0%; p<0.01; Fig 6b). Conversely, the group with an arc angle of >180° (extensive partial arcs) exhibited statistically significant but relatively higher heterogeneity (SMD = −1.48 Gy; 95% CI = −1.80 to −1.15 Gy; I²=40%; p<0.01; Fig 6b), indicating higher effectiveness in the restricted partial arcs group.

The subgroup analysis based on lymph node involvement revealed statistically significant and low heterogeneity in the group with lymph node involvement (SMD = −1.27 Gy; 95% CI = −1.60 to −0.95 Gy; I²=0%; p<0.01; Fig 6c). Conversely,

**Table 5. Risk of bias in included studies.**

| Study ID | Modified New-castle Ottawa Scale | | | | | | | |
|---|---|---|---|---|---|---|---|---|
| | Selection | | | Comparability | Outcome | | Total Score (out of 7) | Power |
| | Representative-ness of exposed cohort(Maxi-mum:★) | Selection of non-exposed cohort(Maxi-mum:★) | Ascertainment of exposure (Maximum:★) | Comparability of cohorts on the basis of the design or analysis(-Maximum: ★★) | Assessment of outcome(-Maximum:★) | Adequacy of follow up of cohorts(Maxi-mum:★) | | |
| Osman 2014 | ★ | ★ | ★ | ★ | ★ | – | ★★★★★(5) | Adequtely powered |
| Swamy 2014 | ★ | ★ | ★ | ★ | ★ | – | ★★★★★(5) | Adequtely powered |
| Pham 2016 | ★ | ★ | ★ | ★ | ★ | – | ★★★★★(5) | Adequtely powered |
| Sakka 2017 | ★ | ★ | ★ | ★ | ★ | – | ★★★★★(5) | Adequtely powered |
| Dumane 2018 | ★ | ★ | ★ | ★ | ★ | – | ★★★★★(5) | Adequtely powered |
| Kuo 2019 | ★ | ★ | ★ | ★ | ★ | – | ★★★★★(5) | Adequtely powered |
| Yeh 2019 | ★ | ★ | ★ | ★ | ★ | – | ★★★★★(5) | Adequtely powered |
| Zhang 2020 | ★ | ★ | ★ | ★ | ★ | – | ★★★★★(5) | Adequtely powered |
| Chen 2020 | ★ | ★ | ★ | ★ | ★ | – | ★★★★★(5) | Adequtely powered |
| Cheung 2022 | ★ | ★ | ★ | ★ | ★ | – | ★★★★★(5) | Adequtely powered |
| Xu 2023 | ★ | ★ | ★ | ★ | ★ | – | ★★★★★(5) | Adequtely powered |

the group without lymph node involvement demonstrated statistically significant but high heterogeneity (SMD = −1.57 Gy; 95% CI = −2.17 to −0.97 Gy; I² = 68%; $p < 0.01$; Fig 6c).

## Publication bias

A funnel plot was used to evaluate publication bias (Fig 7). The result revealed no apparent asymmetry, indicating no obvious publication bias.

## Discussion

RT is a cornerstone in treating residual breast cancer postoperatively and can significantly improve long-term survival [33]. Meta-analyses by the Early Breast Cancer Trialists' Collaborative Group demonstrated prolonged survival benefits, even beyond 15 years, following postoperative radiotherapy. However, radiotherapy induces early and late complications in adjacent tissues, such as the heart and lungs [34].

Radiation-related heart disease (RRHD) is a major concern associated with RT. Thus, reducing heart dose is crucial for minimizing RRHD. This analysis revealed that DIBH yielded significantly lower heart Dmean than FB. Heart V5, V10, V20, V25, and V30 also demonstrated significant reductions. These results indicate the potential of DIBH in mitigating RRHD, potentially lowering risks like coronary artery disease and myocarditis. Radiation pneumonitis (RP), an acute lung injury caused by radiation, poses a significant challenge in thoracic RT. RP occurrence, which is influenced by lung volume and radiation dose parameters, affects treatment plans and patient prognosis. V20 and lung Dmean are key predictors of RP,

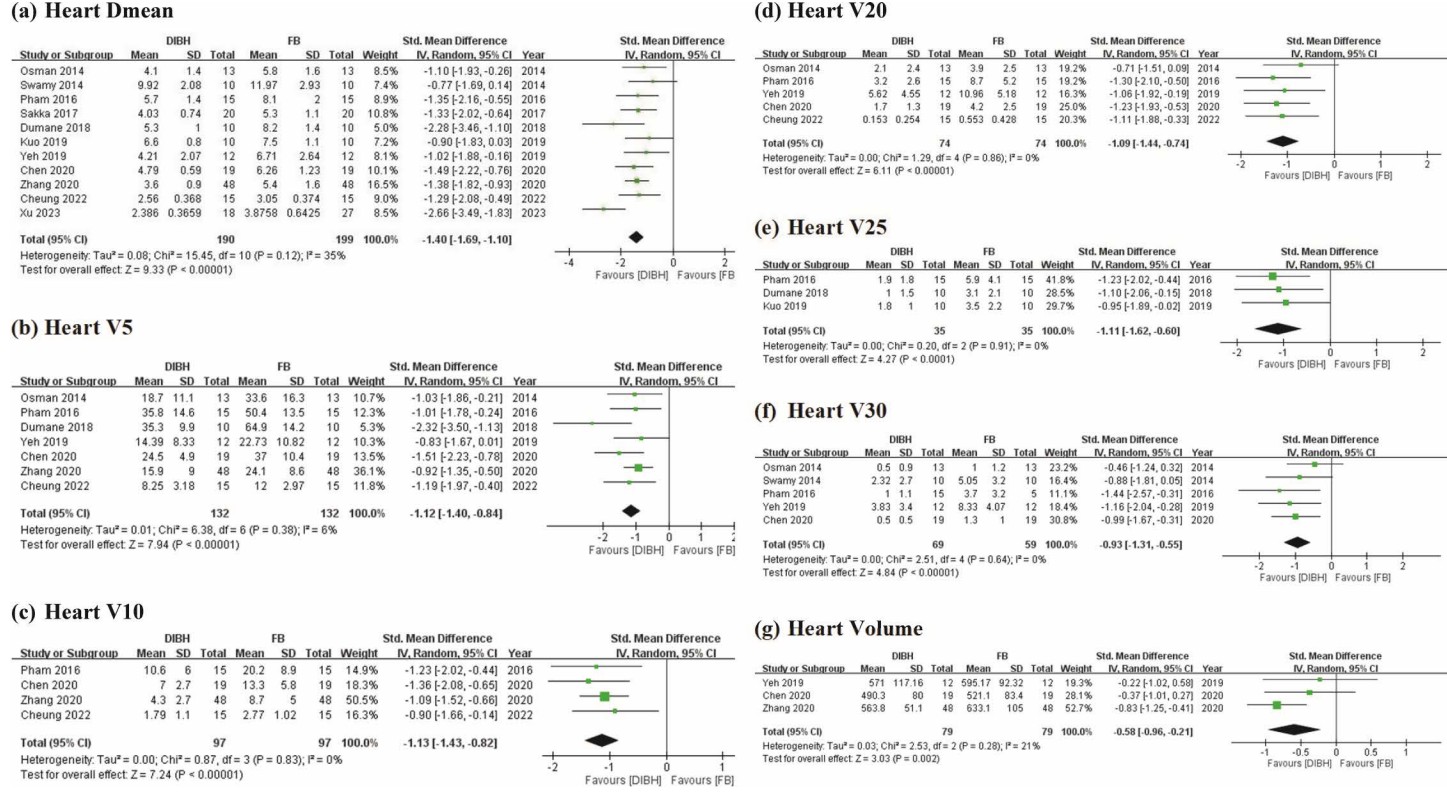

**Fig 2. Forest plots of comparisons.** DIBH vs. FB in postoperative RT strategy for LSBC dose-volume metrics of heart dose (a) Heart Dmean (Gy) (b) Heart V5 (%) (c) Heart V10 (%) (d) Heart V20 (%) (e) Heart V25 (%) (f) Heart V30 (%) and (g) Heart Volume (cc).

predominantly used in clinical practice. Limiting V20 to ≤30% reduces RP and preserves lung function [35]. Other predictors of RP include V5, V25, and V30, which indicate lung volumes exposed to specific radiation doses.

This meta-analysis revealed that the DIBH group demonstrated a lower ipsilateral lung Dmean than the FB group, but with limited strength of evidence due to high heterogeneity. Similar results were observed for ipsilateral lung V5 and V10. However, there was a significant reduction in the V20 for the DIBH group compared with the FB group, with low heterogeneity. During FB, the left lung's proximity to the treatment field may cause higher radiation exposure. Chest and lung movements cause dose variations, potentially resulting in left lung overexposure. DIBH typically reduces the left lung Dmean compared with FB. The heart and left lung move away from the treatment field by requesting patients hold their breath in deep inspiration, thereby improving breast tissue targeting and reducing left lung radiation exposure. Most studies conducted with 3D-CRT or tangential IMRT have revealed that DIBH can reduce Dmean, V5, V10, and V20 in the left lung, although some have observed no significant difference [10,36–38]. Patient anatomy and geometry may play a dominant role in identifying the results. Among the contralateral lung doses, the DIBH group demonstrated a significantly lower Dmean than the FB group. The subgroup analysis of contralateral lung dose V5 revealed similar results, although V20 did not reach statistical significance.

Swamy et al. revealed that the heightened contralateral breast dose cannot be ignored for young patients (<40 years old) in terms of the contralateral breast, as the risk of radiation-induced secondary cancers is increased for doses of >1 Gy. An increased volume of the right lung and right breast was exposed to low doses in the VMAT plans. The low dose–volume was usually greater in VMAT due to multiple beam directions passing through regions outside the planning target

## (a) LAD Dmax

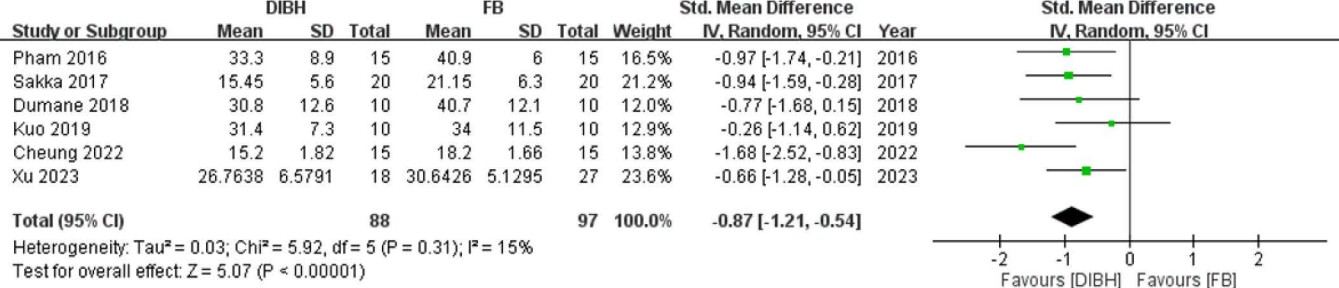

## (b) LAD Dmeans

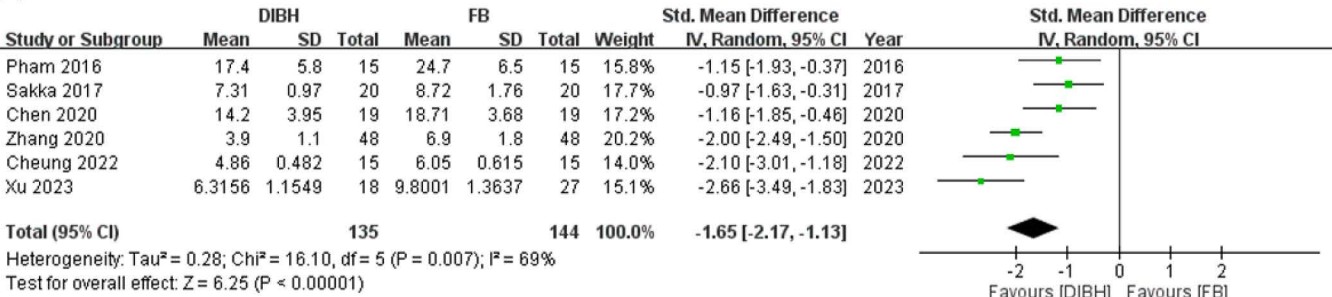

**Fig 3. Forest plots of comparisons.** DIBH vs. FB in postoperative RT strategy for LSBC dose of LAD (a) LAD Dmax (Gy) (b) LAD Dmean (Gy).

volume. The issue of the volume of low doses in the contralateral breast and lung remains unknown for the potential risk of secondary cancers. In this meta-analysis, the forest plot indicated a significantly lower Dmean in the contralateral breast for the DIBH group than for the FB group because patients with DIBH showed increased contralateral lung volume and movement of the contralateral breast away from the treatment beams. This indicates a reduced risk of contralateral breast toxicity or the development of precancerous lesions. However, no statistically significant difference was found in contralateral breast V5.

Previous studies indicated that tumor bed boost, arc angle, and lymph node involvement may influence the heart dose [21]. However, our subgroup analysis revealed that across all subgroups—i.e., breast only, with or without a tumor bed boost, and with or without nodal irradiation—VMAT combined with DIBH consistently resulted in reduced heart Dmean. This strategy is especially important for patients suffering from LSBC with a tumor bed boost or without nodal irradiation. These findings suggest that the cardiac-sparing benefits of DIBH are preserved regardless of common treatment-related factors that have previously been associated with increased heart exposure. One possible explanation is that VMAT's superior dose conformity and arc modulation flexibility enable precise avoidance of critical structures, thereby mitigating the influence of these factors. This highlights the robustness of combining VMAT with DIBH as a planning approach. Our results demonstrate that DIBH is more effective than FB in reducing the heart Dmean, and importantly, this benefit remains stable across different clinical scenarios. This consistent performance across subgroups underscores the clinical value of incorporating DIBH into VMAT-based radiotherapy for LSBC.

Radiotherapy with the arc angle of >180° (extensive partial arcs) demonstrated better heart Dmean reduction between the DIBH and FB groups than RT with the arc angle of <180° (restricted partial arcs). Treatment plans using restricted partial arcs may reduce doses in the gantry incident direction compared with those with extensive partial arcs. However, restricted partial arcs may be necessary to maintain target coverage, compromising the benefit of low-dose exposure. Konstantinou et al. revealed low doses in the LAD and left ventricle Dmean in plans with four restricted partial arcs vs.

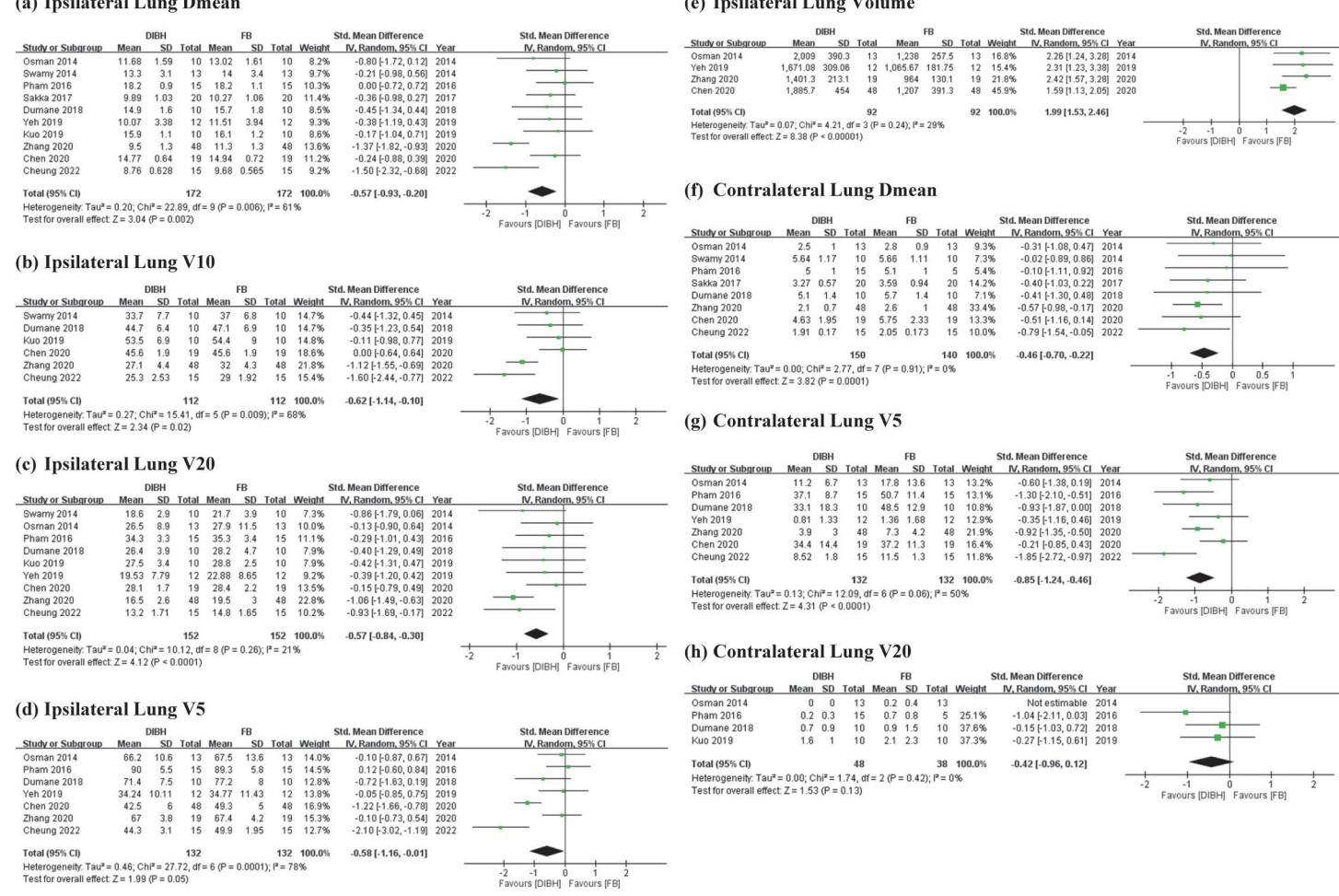

**Fig 4. Forest plots of comparisons.** DIBH vs. FB in postoperative RT strategy for LSBC dose-volume metrics of lungdose **(a)** Ipsilateral Lung Dmean (Gy) **(b)** Ipsilateral Lung V10 (%) **(c)** Ipsilateral Lung V20 (%) **(d)** Ipsilateral Lung V5 (%) **(e)** Ipsilateral Lung Volume (cc) **(f)** Contralateral Lung Dmean (Gy) **(g)** Contralateral Lung V5 (%) **(h)** Contralateral Lung V20 (%).

two extensive partial arcs. Plans with four restricted partial arcs demonstrated slightly higher heart Dmean, V20, and V40, but the differences were insignificant [39]. However, the results regarding which arc angle results in lower doses to OARs when combined with DIBH remain difficult to interpret. These subgroups may confound each other; for example, a different arc angle might be applied in the case of a tumor bed boost, thereby confounding the results.

This study has several limitations that should be considered. First, although a funnel plot analysis suggested no apparent publication bias, the language restriction to English and the use of a limited number of databases may have constrained the comprehensiveness of the literature search. Second, all included studies were retrospective dosimetric comparisons or case series without prospective clinical outcome data or follow-up, which limits the ability to infer long-term clinical benefits. Third, the relatively small sample sizes and single-institution origins of most studies may affect the generalizability of the findings. Additionally, variations in patient anatomy may have influenced the dose received by OARs, introducing heterogeneity and potential bias. Moreover, differences in patient breathing patterns—such as abdominal (diaphragmatic) versus thoracic breathing—may further affect the extent of organ displacement during DIBH. Prior research has shown that abdominal breathing expands the lungs and compresses the heart more effectively, potentially

## (a) Contralateral Breast Dmean

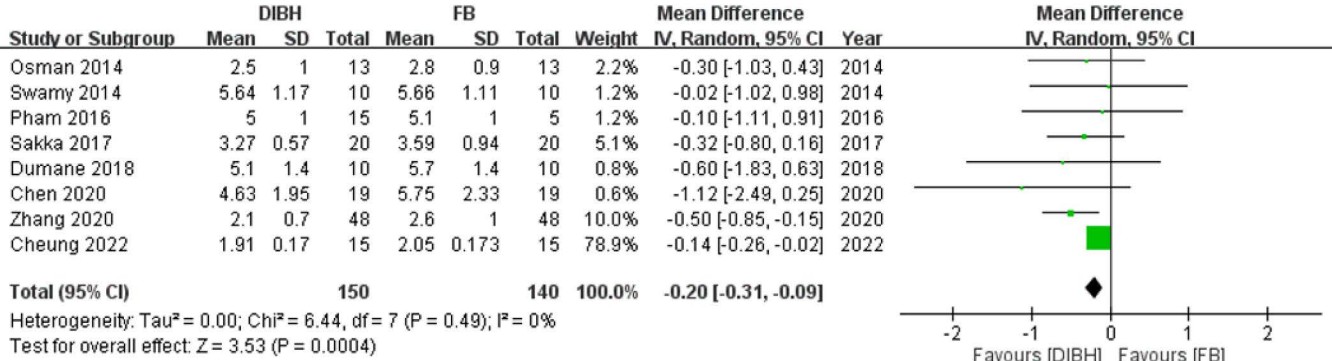

## (b) Contralateral Breast V5

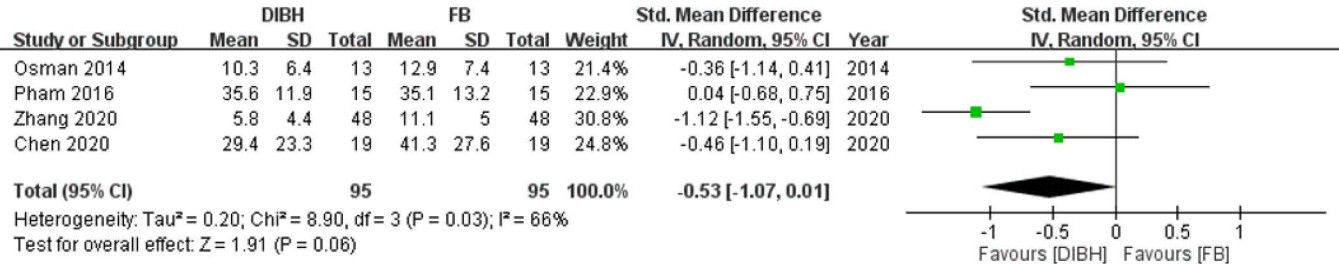

**Fig 5. Forest plots of comparisons.** DIBH vs. FB in postoperative RT strategy for LSBC dose-volume metrics of contralateral breast dose **(a)** Contralateral Breast Dmean (Gy) **(b)** Contralateral Breast V5 (%).

enhancing dosimetric benefits [40]. However, none of the included studies reported or stratified patients based on their breathing type, which could contribute to inter-study variability. Lastly, although prior meta-analyses have compared DIBH and FB across different radiotherapy techniques, this study offers a more focused and detailed assessment of DIBH specifically within the context of VMAT, providing nuanced insights into arc selection and target delineation that are not addressed in broader analyses. These factors should be taken into account when interpreting the conclusions.

In addition to selecting an appropriate breath-hold technique and monitoring system, patient training and staff guidance play a critical role in the successful implementation of DIBH. The ESTRO-ACROP guideline recommends introducing patients to the breath-hold process prior to CT simulation to enhance familiarity and comfort [41]. This can be achieved either directly in the CT suite or through dedicated coaching sessions, especially in centers that are new to implementing DIBH. During coaching, staff should clearly explain the expected breath-hold level—typically a moderately deep inspiration (approximately 70–85% of the patient's maximum capacity)—and the required duration, usually 20–30 seconds. This approach helps prevent breath-hold attempts that are too deep or prolonged, which can be unsustainable and lead to variability during treatment. Consistent verbal cues, visual aids, and instructional materials can improve communication and patient compliance. Identifying whether a patient breathes using the diaphragm or chest may also enhance treatment precision, as thoracic breathing can limit lung expansion and reduce the effectiveness of DIBH.

## Conclusions

This meta-analysis demonstrates that combining DIBH with VMAT in postoperative radiotherapy for LSBC significantly reduces heart Dmean across various clinical scenarios, including subgroups with or without a tumor bed boost and nodal irradiation. DIBH also contributes to lowering radiation exposure to the ipsilateral lung and contralateral breast, suggesting

**(a) Tumor Bed Boost**

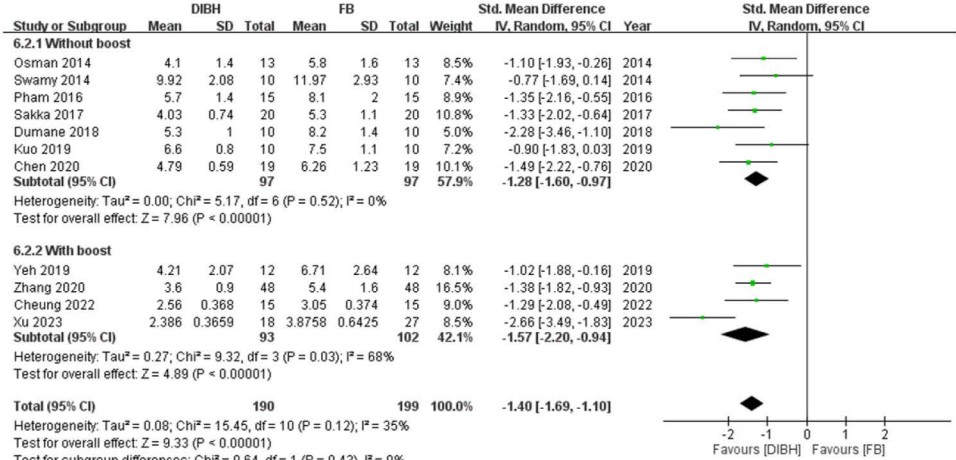

**(b) Arc Angle**

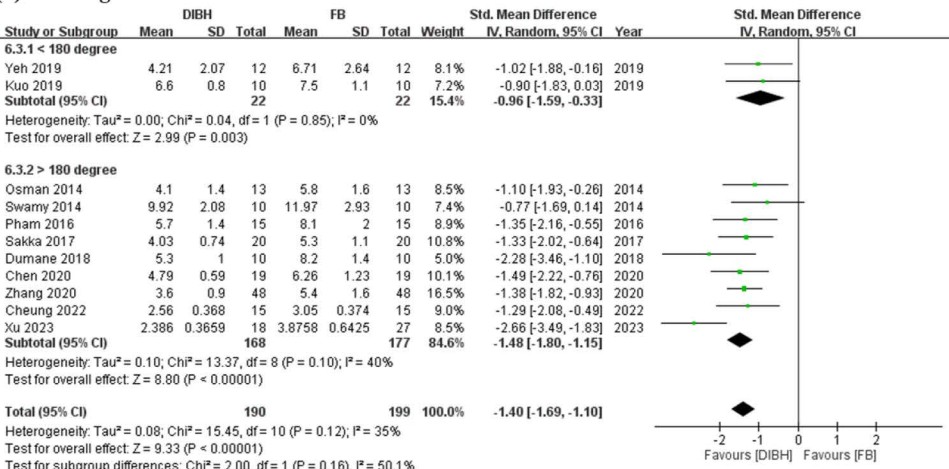

**(c) Lymph Node Involvement**

**Fig 6. Forest plots of comparisons.** DIBH vs. FB in postoperative RT strategy for LSBC of mean heart dose (Gy) (a) Tumor Bed Boost (b) Arc Angle (c) Lymph Node Involvement.

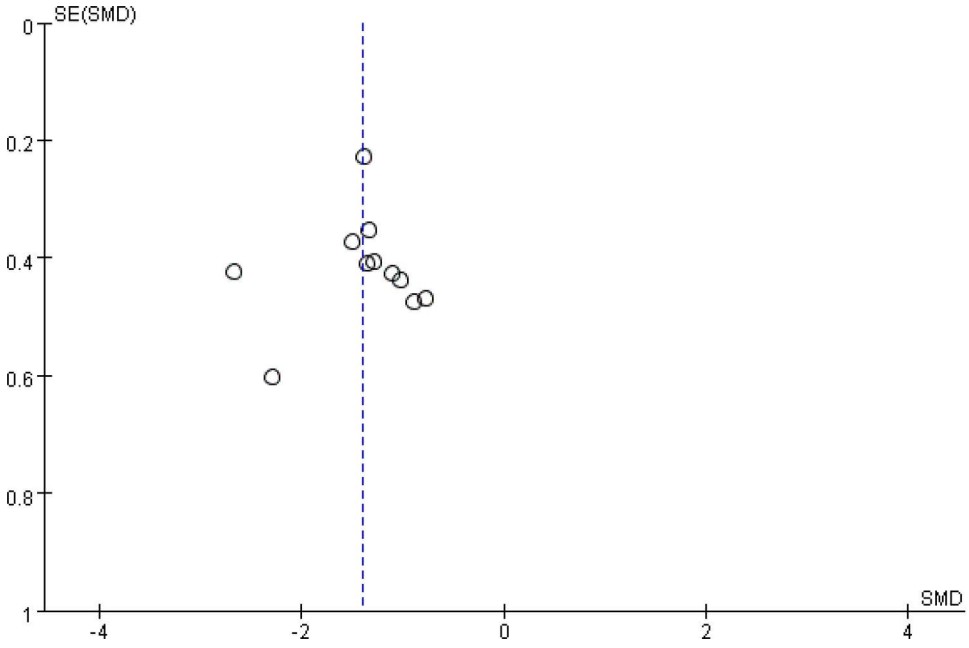

**Fig 7. Funnel plot for publication bias.**

broader organ protection. Despite the inherent increase in low-dose exposure with VMAT, its combination with DIBH still offers a clear dosimetric advantage, particularly in minimizing heart dose. These findings support the routine integration of DIBH in VMAT-based treatment for LSBC and highlight the need for future prospective studies to evaluate long-term outcomes and assess patient-specific factors such as breathing patterns and coaching strategies.

## Protocol registration number

Inplasy, the International Platform of Registered Systematic Review and Meta-analysis Protocols, had 288 approved this study with certification number INPLASY202440043.

## Supporting information

**S1 Checklist. PRISMA 2020 Checklist 0816.**
(DOCX)

## Author contributions

**Conceptualization:** Pin-Yi Chiang, Ching-Po Lin, Chih-Chia Chang.

**Data curation:** Pin-Yi Chiang, Chao-Hsiung Hung.

**Formal analysis:** Pin-Jui Huang.

**Investigation:** Pin-Jui Huang.

**Methodology:** Pin-Jui Huang.

**Project administration:** Chih-Chia Chang.

**Resources:** Ching-Po Lin.

**Software:** Pin-Jui Huang.

**Supervision:** Ching-Po Lin.

**Validation:** Chao-Hsiung Hung.

**Writing – original draft:** Pin-Yi Chiang, Pin-Jui Huang.

**Writing – review & editing:** Ching-Po Lin, Chih-Chia Chang.

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
