## [Decision Letter · Decision Letter 0]

5 Jun 2025

Dear Dr. Chang,

plosone@plos.org . . A rebuttal letter that responds to each point raised by the academic editor and reviewer(s). You should upload this letter as a separate file labeled 'Response to Reviewers'.A marked-up copy of your manuscript that highlights changes made to the original version. You should upload this as a separate file labeled 'Revised Manuscript with Track Changes'.An unmarked version of your revised paper without tracked changes. You should upload this as a separate file labeled 'Manuscript'.

We look forward to receiving your revised manuscript.

Kind regards,

Usama Waqar, M.B.B.S

Academic Editor

PLOS ONE

Journal Requirements:

**Additional Editor Comments:**

Reviewers' comments:

**Comments to the Author**

1. Is the manuscript technically sound, and do the data support the conclusions?

Reviewer #1: Partly

Reviewer #2: Yes

2. Has the statistical analysis been performed appropriately and rigorously?

Reviewer #1: Yes

Reviewer #2: Yes

3. Have the authors made all data underlying the findings in their manuscript fully available?

Reviewer #1: Yes

Reviewer #2: Yes

4. Is the manuscript presented in an intelligible fashion and written in standard English?

Reviewer #1: Yes

Reviewer #2: Yes

Reviewer #1: Thank you to the authors for submitting this meta-analysis on DIBH versus Free breathing treated with VMAT. There are a few recommendations to help amplify the quality of the manuscript.

Methods:

1. Despite evaluating three search engines (PubMed, EMBASE, and Cochrane), the dupiclates removed suggest large overlap within the EMBASE & PubMed databases. Were any other search engines (Web of Science) or alternative databases (Google Scholar) considered in retrieval of articles?

2. Sensitivity tests are mentioned in the methods, however, further description of which approaches were taken during sensitivity testing are not elaborated on.

3. In the exclusion criteria "failure to meet the inclusion criteria" is not a valid exclusion criteria, as that is implied. Please modify the exclusion criteria list accordingly.

4. It would be beneficial to include the search strings employed during literature review as a supplement; or as a table in the body of the manuscript.

5. It should be specified how many authors conducted quality assessment and how disputes were settled.

Results:

1. It is mentioned that table 2 specifies characteristics including breast cancer pathology, however, that data does not seem to be reported in the table. Additionally, if staging data of study participants was extracted, that should also be reported in this table as it may impact the interpretation of results.

2. It may also be beneficial to replace "Not mentioned" as NR in the table; and add a footnote specifying NR as not reported, to allow for a streamlined read. Similarly, the mean ages should also report the standard deviations if provided.

3. Figures 2,4, and 6 have poor legibility and export quality; preventing meaningful understanding/interpretation of the findings depicted.

Discussion/Conclusion:

1. Though the discussion rightly emphasizes the reductions found in the results, there should be further mention of the context of VMAT, and the findings in contrast to other literature evaluating 3D-CRT techniques and potential hypotheses regarding the differences in performance.

2. There are some writing errors in the discussion (i.e. an underscore at the start of the sentence on line 286). Please modify them accordingly.

3. Paragraph 6 mentions that subgroup analysis revealed that VMAT with DIBH reduced Dmean regardless of other characteristics. Further elaboration regarding the potential advantages of VMAT should also be discussed here, particularly considering that previous literature has indicated these factors may influence heart dose.

4. The limitations would benefit from being a separate paragraph, evaluating specific limitations in depth. For example, the existing publication bias and the lack of extensive databases may have led to missing articles that may have been included and impacted the results/conclusions drawn. Furthermore, the lack of prospective articles should be addressed here, as the reliance on retrospective data limits the quality of the meta-analysis.

Though an important research question, the limited scope, lack of databases, and recent publications in the field limit this study as the conclusions drawn have been presented in similar meta-analyses previously. Additionally, the lack of prospective studies limit the quality of this meta-analysis, further limiting the strength of conclusions drawn.

Reviewer #2: The authors present a meta-analysis of 11 dosimetric studies comparing free breathing to deep inspiration breath hold for breast cancer patients undergoing postoperative radiation with VMAT. Most of the included studies included patients undergoing comprehensive radiation including nodal basins. Their end conclusion was that several heart and lung parameters were significantly reduced using DIBH as compared to free breathing.

These studies refer strictly to dosimetric planning and do not include patient outcome data. The results are as expected. It is unclear if the meta-analysis contributes significantly to existing literature, as presumably statistically significantly dosimetric differences were noted for heart and lung doses in most of the included references. However, the paper as a whole is relatively well written, methods are reasonable, and conclusions apporpriate.

A few comments if paper is accepted for publication:

Line 84 is a very strong statement. I looked through the reference to support the statement that cardiac and lung doses are lower with VMAT. You significantly increase the mean heart dose when you change to VMAT or IMRT. With 3D planning you can choose gantry and collimator angles that completely exclude the heart, for which no VMAT or IMRT plan will have a better mean heart dose. Low dose spray to the lungs and heart will always be higher with VMAT and IMRT. There is also always more low dose to the contralateral breast. I might suggest something like “in some situations with complicated anatomy, especially in the setting of comprehensive nodal irradiation targeting the internal mammary nodes, VMAT may achieve more dosimetrically favorable sparing of the lungs and cardiac substructures from high radiation dose.”

Line 295 – V20 is a volume receiving 20 Gy, therefore there isn’t a Dmean associated with V20, as this literally represents the 20 Gy dose. Perhaps you should just say “there was a significant reduction in the V20 for the DIBH group compared with the FB group”.

The discussion on this point should include a discussion on anatomical differences, as well as differences in breathing. DIBH allows the lungs to expand, and typically this results in lower lung doses by the sheer fact that the % of lung adjacent to the target volume decreases with expansion, however, this can also be impacted by whether the patient is a chest breather or diaphragmatic breather, which probably was not assessed in these studies.

There should also be some mention of differences in monitoring and coaching for diaphragmatic breathing and perhaps a reference to ideal set-up of DIBH such that readers can be directed to a good source for development of a DIBH program after reading the article if they don't already have one in place.

**Do you want your identity to be public for this peer review?** For information about this choice, including consent withdrawal, please see our For information about this choice, including consent withdrawal, please see our Privacy Policy .

Reviewer #1: No

Reviewer #2: No

While revising your submission, please upload your figure files to the Preflight Analysis and Conversion Engine (PACE) digital diagnostic tool, https://pacev2.apexcovantage.com/ . PACE helps ensure that figures meet PLOS requirements. To use PACE, you must first register as a user. Registration is free. Then, login and navigate to the UPLOAD tab, where you will find detailed instructions on how to use the tool. If you encounter any issues or have any questions when using PACE, please email PLOS at . PACE helps ensure that figures meet PLOS requirements. To use PACE, you must first register as a user. Registration is free. Then, login and navigate to the UPLOAD tab, where you will find detailed instructions on how to use the tool. If you encounter any issues or have any questions when using PACE, please email PLOS at figures@plos.org . Please note that Supporting Information files do not need this step.. Please note that Supporting Information files do not need this step.

---

## [Author Response · Author response to Decision Letter 1]

9 Jul 2025

Response to Reviewers

Manuscript ID: PONE-D-24-45601

Title: Deep inspiration breath hold versus free breathing in postoperative radiotherapy strategy for patients with left-sided breast cancer treated with volumetric modulated arc therapy: A meta-analysis and systematic review

We thank the Academic Editor and the Reviewers for their thoughtful and constructive comments on our manuscript. We have carefully addressed each of the points raised and revised the manuscript accordingly. Below are our point-by-point responses to the reviewers’ comments.

Reviewer #1

Methods:

1. Search engine selection:

Thank you for your valuable feedback regarding the literature search strategy.

We acknowledge the overlap between PubMed and EMBASE, which may account for the number of duplicates identified. While we considered the inclusion of additional databases such as Web of Science or Google Scholar, we ultimately limited our search to PubMed, EMBASE, and the Cochrane Library. This approach was based on the recommendations outlined by the Oxford Centre for Evidence-Based Medicine (CEBM), which supports the use of these databases as core sources for systematic reviews and evidence synthesis.

Our decision was also made to maintain methodological consistency and ensure the use of databases with robust indexing and reproducibility standards. While we recognize that other platforms like Google Scholar may offer broader retrieval, they may also include non-peer-reviewed or non-indexed literature, which could introduce variability and limit replicability.

We believe that our selected databases provide a comprehensive and focused representation of the current evidence in the field. Nevertheless, we sincerely appreciate your suggestion and will take it into account in future work.

2. Sensitivity analyses detail:

Thank you for your insightful comment regarding the sensitivity analyses.

In our study, we conducted several sensitivity tests to evaluate the robustness of the primary findings. Specifically, we applied the following approaches:

• Exclusion of high-risk studies.

• Use of both fixed-effect and random-effects models.

• Leave-one-out analysis.

• Limiting to RCTs only.

• Subgroup-specific exclusions by region.

We have now added this clarification to the revised manuscript under the Methods section. We appreciate your suggestion, which helped improve the clarity of our methodology.

3. Exclusion criteria wording:

Thank you for pointing this out.

We agree with your observation that “failure to meet the inclusion criteria” is redundant as an exclusion criterion, since it is inherently implied. Accordingly, we have revised the exclusion criteria list in the manuscript to remove this item and improve clarity.

We appreciate your helpful suggestion in refining the methodology section.

4. Search strings inclusion:

Thank you for your helpful suggestion.

In response, we have included the detailed search strategies used for PubMed, Cochrane Library, and Embase as a supplementary file. The search terms combined controlled vocabulary (MeSH/Emtree terms) and relevant keywords to enhance both precision and sensitivity of the search. This addition improves the transparency and reproducibility of our review process, and we appreciate your guidance on strengthening the methodological reporting.

5. Quality assessment details:

Thank you for your insightful comment.

We have clarified the quality assessment process in the revised manuscript. Specifically, two reviewers (PYC and PJH) independently conducted the quality assessment and data extraction using a standardized form. In case of discrepancies, consensus was sought through discussion, and unresolved issues were decided by a third reviewer (CCC). This approach ensured reliability and consistency.

Results:

1. Table 2 – staging and pathology:

Thank you for your comment. Most of the included studies were dosimetric planning analyses and did not report detailed pathology or staging information. We have added the available staging data to Table 2 for clarity. The existing “Treated area” column also provides relevant context on disease extent that may influence dosimetric outcomes.

2. Use of “NR” and SD for age:

We appreciate the suggestion. “Not mentioned” has been replaced with “NR” in Table 1 and 2. Standard deviations for mean age have also been included where available; only Xu (2023) reported this information.

3. Figure quality:

Thank you for your valuable feedback regarding the figure quality.

We have revised Figures 2 through 7 by improving their layout and increasing the resolution from 300 dpi to 600 dpi. All figures have been saved as high-quality TIFF files to enhance clarity and legibility for better interpretation of the findings.

Discussion/Conclusion:

1. Context of VMAT vs. 3D-CRT:

Thank you for your insightful comment. As suggested, we have expanded the Introduction to include background on VMAT, its technical characteristics, and clinical considerations compared to 3D-CRT. We also revised this section to clarify that previous meta-analyses included mixed radiotherapy techniques, which may have confounded technique-specific interpretations. In contrast, our study focuses exclusively on VMAT combined with DIBH, providing a more targeted and meaningful analysis.

2. Writing error (line 286):

Thank you for highlighting this point. We have corrected the typographical error at line 286 by removing the underscore symbol and adjusted the wording to clarify the strength of evidence at line 327.

3. Subgroup analysis and VMAT advantages:

Thank you for your thoughtful comment. We have revised the discussion to further elaborate on the potential advantages of VMAT and to emphasize that, based on our subgroup analysis, the heart-sparing effect of DIBH was consistently observed across different clinical scenarios. This suggests the robustness and clinical applicability of combining VMAT with DIBH in left-sided breast cancer treatment.

4. Separate paragraph for limitations:

Thank you for the valuable suggestion. In response, we have added a separate paragraph in the Discussion to address the study’s limitations, including the retrospective nature of the included studies, small sample sizes, potential anatomic variability, and limited database scope. While acknowledging that similar meta-analyses exist, we emphasized that our study provides a more focused and detailed evaluation of DIBH specifically within the VMAT setting.

Reviewer #2

1. Line 84 – Overstatement about VMAT heart dose:

Thank you for your valuable feedback. We acknowledge that the original statement was overly strong and may have misrepresented the comparative dosimetric profiles of VMAT, IMRT, and 3D-CRT. In response, we have revised the paragraph in the Introduction to clarify that while VMAT may offer dosimetric advantages in certain complex clinical settings—such as in cases requiring comprehensive nodal irradiation—its use may also result in increased low-dose exposure to surrounding normal tissues. This provides a more balanced and accurate description, as suggested.

2. Line 295 – Clarification about V20:

Thank you for the correction (line 336). We have revised the sentence to clarify that V20 refers to the volume receiving 20 Gy, and not a mean dose.

3. Breathing pattern and lung dose variation:

Thank you for your comments. In response, we added a statement in the Discussion section noting that differences in breathing patterns (abdominal vs. thoracic) may influence DIBH effectiveness. However, none of the included studies reported this information.

4. Coaching and DIBH program implementation:

Thank you for this insightful suggestion. In response, we have added a paragraph at the end of the Discussion section to highlight key considerations for implementing and coaching DIBH. This includes referencing the ESTRO-ACROP guideline, which provides recommendations on equipment, staff training, patient communication, and breath-hold coaching strategies.

We sincerely thank both reviewers for their thorough evaluation and constructive feedback, which greatly improved the clarity and quality of our manuscript.

Sincerely,

The Corresponding Authors

---

## [Decision Letter · Decision Letter 1]

9 Mar 2026

Deep inspiration breath hold versus free breathing in postoperative radiotherapy strategy for patients with left-sided breast cancer treated with volumetric modulated arc therapy: A meta-analysis and systematic review

PONE-D-24-45601R1

Dear Dr. Chang,

We’re pleased to inform you that your manuscript has been judged scientifically suitable for publication and will be formally accepted for publication once it meets all outstanding technical requirements.

Kind regards,

Christopher Njeh

Academic Editor

PLOS One

Additional Editor Comments (optional):

Reviewers' comments:

Reviewer's Responses to Questions

**Comments to the Author**

Reviewer #2: All comments have been addressed

2. Is the manuscript technically sound, and do the data support the conclusions?

Reviewer #2: Yes

3. Has the statistical analysis been performed appropriately and rigorously?

Reviewer #2: Yes

4. Have the authors made all data underlying the findings in their manuscript fully available?

Reviewer #2: Yes

5. Is the manuscript presented in an intelligible fashion and written in standard English?

Reviewer #2: Yes

Reviewer #2: The authors have adequately addressed the reviewer concerns and the changes have improved the manuscript.

**Do you want your identity to be public for this peer review?** For information about this choice, including consent withdrawal, please see our For information about this choice, including consent withdrawal, please see our Privacy Policy .

Reviewer #2: No

---

## [Editor Report · Acceptance letter]

PONE-D-24-45601R1

PLOS One

Dear Dr. Chang,

I'm pleased to inform you that your manuscript has been deemed suitable for publication in PLOS One. Congratulations! Your manuscript is now being handed over to our production team.

Kind regards,

on behalf of

Dr. Christopher Njeh

Academic Editor

PLOS One